# LAVITA: LATENT VIDEO DIFFUSION MODELS WITH SPATIO-TEMPORAL TRANSFORMERS

## ABSTRACT

Video generation is a challenging task as it requires effective modeling of rich spatio-temporal information from high-dimensional video data. To tackle this challenge, we propose a novel architecture, the **la**tent **vi**deo diffusion model with spatio-temporal **tra**nsformers, referred to as LAVITA, which integrates the Transformer architecture into diffusion models for the first time within the realm of video generation. Conceptually, LATIVA models spatial and temporal information separately to accommodate their inherent disparities as well as to reduce the computational complexity. Following this design strategy, we design several Transformer-based model variants to integrate spatial and temporal information harmoniously. Moreover, we identify the best practices in architectural choices and learning strategies for LAVITA through rigorous empirical analysis. Our comprehensive evaluation demonstrates that LAVITA achieves state-of-the-art performance across several standard video generation benchmarks, including FaceForensics, SkyTimelapse, UCF101, and Taichi-HD, outperforming current best models. We strongly believe that LAVITA provides valuable insights for future research on incorporating Transformers into diffusion models for video generation.

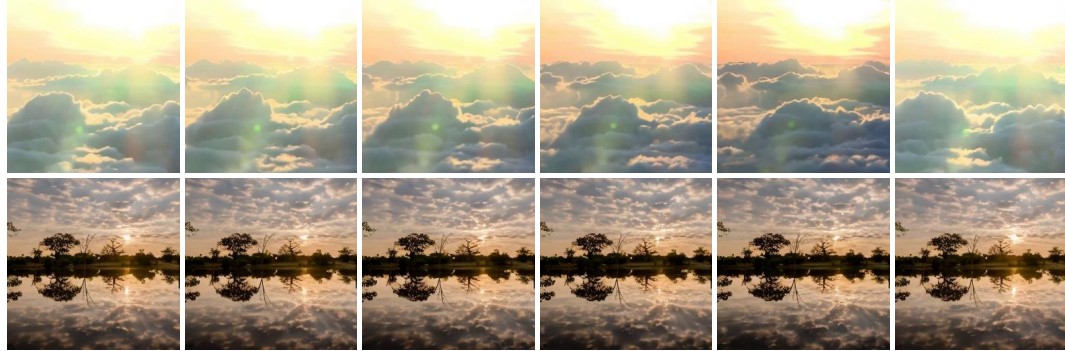

Figure 1: Sample videos ($256 \times 256$, starting from the first frame and displaying every two frames). LAVITA generates photorealistic videos with temporal coherent content.

## 1 INTRODUCTION

Diffusion models (Ho et al., 2020; Song et al., 2021) are the powerful deep generative model that has emerged recently for various tasks, including image-to-image generation (Meng et al., 2022; Zhao et al., 2022), text-to-image generation (Zhou et al., 2023; Rombach et al., 2022), and 3D scenes (Wang et al., 2023; Chen et al., 2023), etc. Compared to these successful applications in images, generating high-quality videos still faces significant challenges, which can be primarily attributed to the intricate and high-dimensional nature of videos that encompass complex spatio-temporal dynamics within high-resolution frames (Yu et al., 2022).

Simultaneously, researchers have unveiled the significance of revolutionizing backbones in the success of diffusion models (Nichol & Dhariwal, 2021; Peebles & Xie, 2022; Bao et al., 2023). The

U-Net (Ronneberger et al., 2015), which relies on convolutional neural networks (CNNs), has held a prominent position in existing works (Ho et al., 2022; Dhariwal & Nichol, 2021). Conversely, the Vision Transformer (ViT) (Dosovitskiy et al., 2021) has shown that architectures centered entirely on Transformers (Vaswani et al., 2017) can outperform convolutional structures in the realm of image classification. Moreover, Peebles & Xie (2022) have demonstrated that the inductive bias of U-Net is not crucial for the performance of diffusion models. In contrast, attention-based architectures present an intuitive option for capturing long-range contextual relationships in videos. Therefore, a very natural question arises: *Can Transformer-based diffusion models enhance the generation of realistic videos?*

Two primary challenges need to be addressed to answer this question. Firstly, videos contain a substantial number of tokens, posing the question of how to effectively model the spatio-temporal information. Considering the inherent disparities between spatial and temporal information, it is natural to separately model spatial and temporal elements. Secondly, this separation strategy presents us with another two problems of how to independently model temporal and spatial information and subsequently harmonize these two disparate information. Drawing inspiration from the success of Transformer-based models in video classification tasks (Arnab et al., 2021), we adopt a design strategy that spans from the broader perspective of Transformer blocks to the finer details of multi-head attention mechanisms within Transformer blocks. This results in the proposition of four Transformer-based models tailored for spatio-temporal information modeling. Finally, we develop a simple and general **la**tent **vi**deo diffusion model incorporating spatio-temporal **tra**nsformers called LAVITA. This pioneering model marks the first integration of the Transformers into latent diffusion models for video generation.

Moreover, convolutional models have been extensively developed within the community over several years, leading to the establishment of many "best practices" associated with these models (Arnab et al., 2021). Nevertheless, transformer-based latent diffusion models for video generation might demonstrate different characteristics, necessitating the identification of optimal design choices for this architecture. To address this, we conduct a comprehensive ablation analysis encompassing video clip patch embedding, conditional information injection, model architecture, temporal positional embedding, and learning strategies. Our analysis enables LAVITA to generate photorealistic videos with temporal coherent content (see Fig. 1) and achieve state-of-the-art performance across multiple standard video generation benchmarks, including FaceForensics (Rössler et al., 2018), Sky-Timelapse (Xiong et al., 2018), UCF101 (Soomro et al., 2012) and Taichi-HD (Siarohin et al., 2019). Remarkably, LAVITA substantially outperforms all current best models, achieving the best FVD, FID and Inception Score on all evaluated datasets. Moreover, LAVITA is also capable of text-to-video generation, and our evaluation on Webvid2M produces natural and coherent videos given textual prompts. We firmly believe that LAVITA offers valuable insights for future research regarding the utilization of Transformer-based backbones in diffusion models for video generation.

## 2 RELATED WORK

**Video generation** aims to produce realistic videos that exhibit a high-quality visual appearance and consistent motion simultaneously. Previous research in this field can be categorized into three main categories. Firstly, several studies have sought to extend the capabilities of powerful GAN-based image generators to create videos (Vondrick et al., 2016; Saito et al., 2017; Wang et al., 2020; Kahembwe & Ramamoorthy, 2020). However, these methods often encounter challenges related to mode collapse, limiting their effectiveness. Secondly, some methods propose learning the data distribution using autoregressive Transformer models (Ge et al., 2022; Rakhimov et al., 2021; Weissenborn et al., 2020; Yan et al., 2021). While these approaches generally offer better video quality compared to GAN-based methods and exhibit more stable convergence, they come with the drawback of requiring significant computational resources. Finally, recent advances in video generation have focused on building systems based on diffusion models (Ho et al., 2020; Harvey et al., 2022; Ho et al., 2022; Mei & Patel, 2023), resulting in promising outcomes. However, Transformer-based diffusion models have not been well explored.

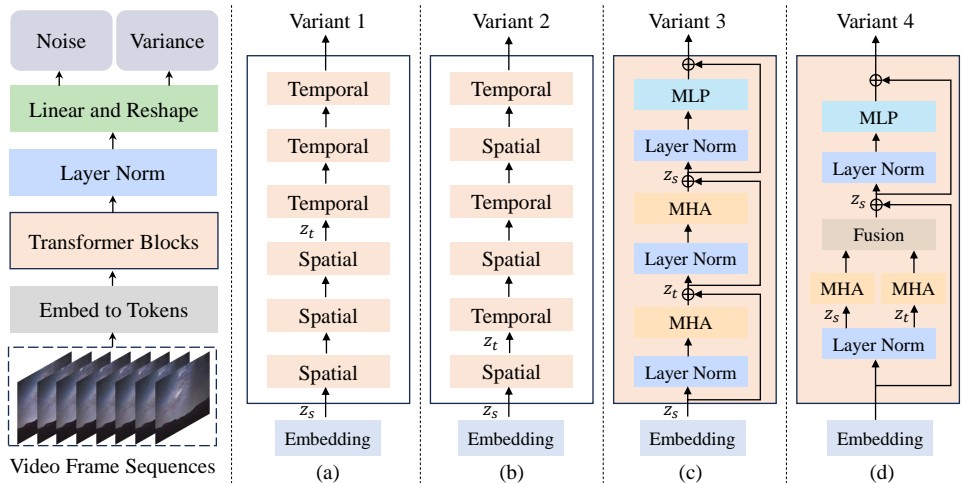

Figure 2: **The architecture of LAVITA for video generation**. We propose several variants of LAVITA to process a large number of spatio-temporal tokens effectively. Each block depicted in light orange represents a Transformer block. (a) and (b) employ the standard Transformer block described in Appx. A.2. Meanwhile, (c) and (d) employ our respective Transformer block variants.

## 3 METHODOLOGY

### 3.1 PRELIMINARY OF LATENT DIFFUSION MODELS

**Latent diffusion model (LDM)** (Rombach et al., 2022). LDM represents an efficient version of the diffusion model (Ho et al., 2020; Song et al., 2021) by conducting the diffusion process in the latent space instead of the pixel space. LDM initiates this process by employing an encoder $\mathcal{E}$ from a pre-trained variational autoencoder to compress the input data sample $x \in p_{\text{data}}(x)$ into a lower-dimensional latent code $z = \mathcal{E}(x)$. Subsequently, it learns the data distribution through two key processes: diffusion and denoising.

The diffusion process gradually introduces Gaussian noise into the latent code $z$, generating a perturbed sample $z_t = \sqrt{\overline{\alpha}_t} z + \sqrt{1 - \overline{\alpha}_t} \epsilon$, where $\epsilon \sim \mathcal{N}(0, 1)$, following a Markov chain spanning $T$ stages. In this context, $\overline{\alpha}_t$ serves as a noise scheduler, with $t$ representing the diffusion timestep.

The denoising process is trained to understand the inverse diffusion process to predict a less noisy $z_{t-1}$: $p_\theta(z_{t-1}|z_t) = \mathcal{N}(\mu_\theta(z_t), \Sigma_\theta(z_t))$ with the variational lower bound of log-likelihood reducing to $\mathcal{L}_\theta = -\log p(z_0|z_1) + \sum_t D_{KL}((q(z_{t-1}|z_t, z_0)||p_\theta(z_{t-1}|z_t)))$. Here, $\mu_\theta$ is implemented using a denoising model $\epsilon_\theta$ and is trained with the *simple* objective,

$$\mathcal{L}_{simple} = \mathbb{E}_{\mathbf{z} \sim p(z), \, \epsilon \sim \mathcal{N}(0,1), \, t} \left[ \|\epsilon - \epsilon_\theta(\mathbf{z}_t, t)\|_2^2 \right]. \tag{1}$$

In accordance with Nichol & Dhariwal (2021), to train diffusion models with a learned reverse process covariance $\Sigma_\theta$, it is necessary to optimize the full $D_{KL}$ term and thus train with the full $\mathcal{L}$, denoted as $\mathcal{L}_{vlb}$. Additionally, $\Sigma_\theta$ is implemented using $\epsilon_\theta$. Fig. 2 (left) illustrates our LAVITA architecture, which integrates Transformers into LDM. Here, $\epsilon_\theta$ is realized with a Transformer.

### 3.2 LAVITA MODEL VARIANTS FOR VIDEO GENERATION

As depicted in Fig. 2, we design four variants of LAVITA utilizing a top-down design strategy that spans from the broader perspective of Transformer blocks to the finer details of multi-head attention mechanisms within Transformer blocks.

**Variant 1.** As shown in Fig. 2 (a), taking inspiration from ViViT (Arnab et al., 2021), this variant comprises two distinct types of Transformer blocks: spatial Transformer blocks and temporal Transformer blocks. The former focuses on capturing spatial information exclusively among tokens sharing the same temporal index, while the latter captures temporal information among tokens extracted from different temporal indices.

In contrast to video or image understanding tasks (Arnab et al., 2021), this variant avoids the incorporation of an extra classification token. Furthermore, we do not employ global average pooling on tokens produced by the spatial Transformer block, such that the tokens are kept their original temporal dimensions, denoted as $z_s \in \mathbb{R}^{n_f \times t \times d}$. Here, $t = n_h \times n_w$ signifies the token count of each temporal index. We reshape $z_s$ into $z_t \in \mathbb{R}^{t \times n_f \times d}$ and feed it into the subsequent temporal Transformer block, which captures temporal information among tokens extracted from different temporal indexes.

**Variant 2.** In contrast to the temporal "late fusion" design in Variant 1 (Neimark et al., 2021; Simonyan & Zisserman, 2014), this variant utilizes the "interleaved fusion" approach to combine spatio-temporal information. As depicted in Fig. 2 (b), this variant consists of an equal number of Transformer blocks as in Variant 1, with alternating focus on capturing spatial and temporal information. Similar to Variant 1, the input shapes for the spatial Transformer block and temporal Transformer block are $z_s \in \mathbb{R}^{n_f \times t \times d}$ and $z_t \in \mathbb{R}^{t \times n_f \times d}$ respectively.

**Variant 3.** Variants 1 and 2 primarily focus on the factorization of the Transformer blocks. Our next focus will be decomposing the multi-head attention in the Transformer block. Illustrated in Fig. 2 (c), this variant forces the process to initially compute self-attention exclusively across the spatial dimension, followed by the temporal dimension, rather than conducting self-attention across all tokens in $z_s$ or $z_t$. As a result, each Transformer block captures interactions that encompass both spatial and temporal aspects. Similar to the previous variants, the inputs for spatial multi-headed self-attention and temporal multi-headed self-attention are $z_s \in \mathbb{R}^{n_f \times t \times d}$ and $z_t \in \mathbb{R}^{t \times n_f \times d}$, respectively.

**Variant 4.** Within this variant, we decompose the multi-head attention (MHA) into two components, with each component utilizing half of the attention heads as shown in Fig. 2 (d). We employ different components to attend over tokens along spatial and temporal dimensions separately. The input shapes for these different components are $z_s \in \mathbb{R}^{n_f \times t \times d}$ and $z_t \in \mathbb{R}^{t \times n_f \times d}$ respectively. Once two different attention operations are concluded, we reshape $z_t \in \mathbb{R}^{t \times n_f \times d}$ into $z_t^{'} \in \mathbb{R}^{n_f \times t \times d}$. This reshaped $z_t^{'}$ is then added to $z_s$, which is the output of the attention operation along the spatial dimension. Finally, this amalgamated tensor is the input for the multilayer perception (MLP) component in the Transformer block.

After the final Transformer block, a critical procedure involves decoding the video token sequence to derive both predicted noise and predicted covariance. The shape of the two outputs is the same as that of the input video clip $V \in \mathbb{R}^{F \times H \times W \times C}$. Following previous work (Peebles & Xie, 2022; Bao et al., 2023), we accomplish this by employing a standard linear decoder in conjunction with a reshape operation. In the context of using the compression frame patch embedding method, an additional step entails integrating a 3D transposed convolution for temporal upsampling of the output videos, following the standard linear decoder and reshaping operation.

### 3.3 VIDEO CLIP PATCH EMBEDDING

We explore two straightforward variations for translating a video clip $V \in \mathbb{R}^{F \times H \times W \times C}$ to a sequence of tokens, denoted as $\hat{z} \in \mathbb{R}^{n_f \times n_h \times n_w \times d}$. Here $d$ represents the dimension of each token. Spatio-temporal positional embedding will be incorporated into $\hat{z}$. Finally, we get the input $z$ for the Transformer by reshaping $\hat{z}$.

**Uniform frame patch embedding.** As illustrated in Fig. 3 (a), we apply the patch embedding technique outlined in ViT (Dosovitskiy et al., 2021) to each video frame individually and then concatenate all of these tokens. Specifically, $n_f$, $n_h$, and $n_w$ are equivalent to $F$, $\frac{H}{h}$, and $\frac{W}{w}$ when non-overlapping image patches are extracted from every video frame. Here, $h$ and $w$ are the height and weight of the image patch, respectively.

**Compression frame patch embedding.** Another approach is to naturally extend the ViT patch embedding method to 3D, as shown in Fig. 3 (b). We extract tubes along the temporal dimension with a certain stride of $s$ and then map them to tokens. $n_f$ is equivalent to $\frac{F}{s}$ in contrast to non-overlapping uniform frame patch embedding. Compared to the uniform frame patch embedding, this method inherently incorporates spatio-temporal information during the patch embedding stage.

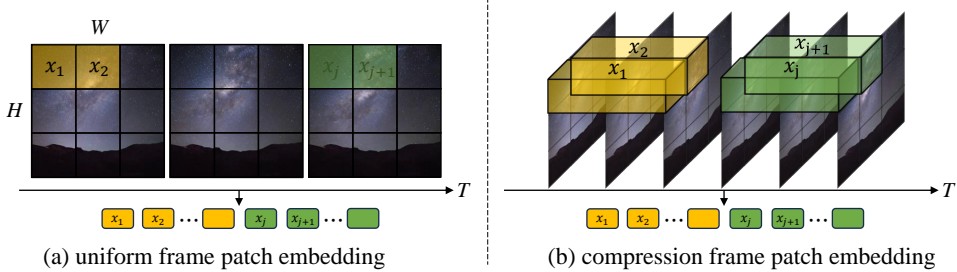

Figure 3: **The video clip patch embedding**. (a) We sample $F$ frames and embed each individual video frame into tokens using the method described in ViT. (b) We extend the ViT patch embedding method from 2D to 3D and subsequently extract tubes along the temporal dimension.

## 3.4 CONDITIONAL INFORMATION INJECTION

We explore two methods for integrating conditional information $c$ (representing either timesteps or labels), into our model. The first approach involves treating it as tokens, and we refer to this approach as *all tokens*. The second method is akin to adaptive layer normalization (AdaLN) (Perez et al., 2018; Peebles & Xie, 2022). We employ linear regression to compute $\gamma_c$ and $\beta_c$ based on the input $c$, resulting in the equation $AdaLN(h, c) = \gamma_c \text{LayerNorm}(h) + \beta_c$, where $h$ represents the hidden embeddings within the Transformer blocks. Furthermore, we also perform regression on $\alpha_c$, which is applied directly before any residual connections (RCs) within the Transformer block, resulting in the equation $RCs(h, c) = \alpha_c h + AdaLN(h, c)$. We refer to this as scalable adaptive layer normalization (*S-AdaLN*). The architecture of *S-AdaLN* can be seen in Appx. A.2.

## 3.5 TEMPORAL POSITIONAL EMBEDDING

We explore two approaches to introduce positional encoding to video tokens along the temporal dimension. We utilize the absolute positional encoding method, incorporating sine and cosine functions with varying frequencies (Vaswani et al., 2017), to enable the model to comprehend the video sequence effectively. Additionally, We also explore the relative positional encoding method by employing rotary positional embedding (RoPE) (Su et al., 2021).

## 3.6 LEARNING TO SYNTHESIZE HIGH-QUALITY VIDEOS

Our goal is to ensure that the generated videos exhibit the best visual quality while preserving temporal consistency. We train all our models following the methodologies outlined in Sec. 3.1, employing both $\mathcal{L}_{simple}$ and $\mathcal{L}_{vlb}$. Building on this foundation, we explore whether the incorporation of two additional learning strategies can enhance both the spatial and temporal quality of the generated videos.

**Learning with pre-trained models.** We utilize the image pre-trained model DiT (Peebles & Xie, 2022) to initialize our video generation model. To address the challenges arising from missing or incompatible parameters during this initialization process, we implement the following strategies. In the DiT pre-trained model, a positional embedding $\boldsymbol{p_s} \in \mathbb{R}^{n_h \times n_w \times d}$ is applied to each token. However, in our video generation model, we have a token count that is $n_f$ times greater. Consequently, we straightforwardly replicate the positional embedding $n_f$ times (Arnab et al., 2021). Furthermore, the DiT pre-trained model includes a label embedding. Nevertheless, the used video dataset either lacks label information or encompasses a significantly smaller number of categories in comparison to ImageNet (Deng et al., 2009). Hence, we opt to directly discard the relative parameters of DiT's label embedding and initialize them with zero values.

**Learning with image-video joint training.** Our video generation model's structure enables simultaneous training for video and image generation by appending randomly independently sampled video frames from the same dataset to the end of the sampled videos (Ho et al., 2022; Blattmann et al., 2023). To ensure our model can generate continuous videos, tokens related to video content are used in the temporal module for modeling temporal information, while image frame tokens are excluded.

## 4 EXPERIMENTS

This section initially outlines the experimental setup, encompassing datasets, evaluation metrics, baselines, and implementation details. Subsequently, we present ablation experiments for each component in our model. Lastly, We then compare experimental results with state-of-the-art methods and present text-to-video generation results.

### 4.1 EXPERIMENTAL SETUP

**Datasets.** We primarily conduct comprehensive experiments on four public datasets: FaceForensics (Rössler et al., 2018), SkyTimelapse (Xiong et al., 2018), UCF101 (Soomro et al., 2012), Taichi-HD (Siarohin et al., 2019). Following the experimental setup in recent video generation works (Skorokhodov et al., 2022), except for UCF101, we use the training split for all datasets when available. For UCF101, we use both the training and testing split. We extract 16-frame video clips from these datasets using a specific sampling interval, with each frame resized to a resolution of 256×256 for training.

**Evaluation metrics.** In the assessment of quantitative comparisons, we employ three evaluation metrics: Fréchet Video Distance (FVD) (Unterthiner et al., 2018), Fréchet Inception Distance (FID) (Parmar et al., 2021), and Inception Score (IS) (Saito et al., 2017). Our primary focus rests on FVD, as it aligns more closely with human subjective judgment and effectively gauges the quality of the generated videos. Adhering to the evaluation guidelines introduced by StyleGAN-V, we compute the FVD scores by analyzing 2,048 video clips, each comprising 16 frames in length. We only employ Inception Score for assessing the generation quality on UCF101, as it leverages the UCF101-fine-tuned C3D model (Saito et al., 2017). All frames within the generated videos are employed to compute the FID metrics.

**Baselines.** We select the following recent and strong methods to evaluate both quantitative and qualitative outcomes, including MoCoGAN (Tulyakov et al., 2018), MoCoGAN with the Style-GAN2 backbone (referred to as MoCoGAN-Style) (Karras et al., 2020), VideoGPT (Yan et al., 2021), MoCoGAN-HD (Tian et al., 2021), DIGAN (Yu et al., 2022), StyleGAN-V (Skorokhodov et al., 2022), and PVDM (Yu et al., 2023). Furthermore, we conduct an extra comparison of IS values between our proposed method and previous approaches on the UCF101 dataset. Unless explicitly stated otherwise, all presented values are obtained from the latest relevant studies: StyleGAN-V and PVDM.

**Implementation details.** We use the AdamW optimizer with a constant learning rate $1 \times 10^{-4}$ to train all models. Horizontal flipping is the only employed data augmentation. Following common practices within generative modeling works (Peebles & Xie, 2022; Bao et al., 2023), an exponential moving average (EMA) of LAVITA weights is upheld throughout training, employing a decay rate of 0.9999. All the reported results exclusively utilize the EMA model.

### 4.2 ABLATION STUDY

In this section, we examine the effect of different designs described in Sec. 3 on the FaceForensics dataset regarding FVD.

**Video clip patch embedding.** We examine the impact of two video clip patch embedding methods detailed in Sec 3.3 when employed with our Variant 2. In Fig. 5 (e), the performance of the compression frame patch embedding methods notably falls behind that of the uniform frame patch embedding method. This finding contradicts the results obtained by the video understanding method ViViT. We speculate that enhancing the temporal upsampling module's capabilities could potentially boost the performance of the previous patch embedding method. Therefore, we select the uniform frame patch embedding method for all subsequent experiments.

**Model variants.** We evaluate our proposed model variants as detailed in Sect 3.2. We strive to equate the parameter counts across all variants to ensure a fair comparison. We commence training all the models from scratch. As shown in Fig. 5 (d), Variant 2 performs the best with increasing iterations. Notably, Variant 4 exhibits roughly a quarter of the floating-point operations (FLOPs) compared to the other three models, as detailed in Appx A.1. Therefore, it is unsurprising that Variant 4 performs the least favorably among the four variants.

| Method | FaceForensics | SkyTimelapse | UCF101 | TaichiHD |
|---|---|---|---|---|
| MoCoGAN | 124.7 | 206.6 | 2886.9 | - |
| MoCoGAN-Style | 55.62 | 85.88 | 1821.4 | - |
| VideoGPT | 185.9 | 222.7 | 2880.6 | - |
| MoCoGAN-HD | 111.8 | 164.1 | 1729.6 | 128.1 |
| DIGAN | 62.5 | 83.11 | 1630.2 | 156.7 |
| StyleGAN-V | 47.41 | 79.52 | 1431.0 | - |
| PVDM | 355.92 | 436.62 | 2064.0 | 540.2 |
| LAVITA (ours) | 34.00 | 59.82 | 863.41 | 159.60 |
| LAVITA+IMG (ours) | **27.08** | **42.67** | **800.54** | **116.83** |

Table 2: **FVD values of video generation models on different datasets**. FVD values for other baseline models are reported and sourced from the reference StyleGAN-V. Additionally, we re-train PVDM using the official implementation.

In Variant 1, half of the Transformer blocks are initially employed for spatial modeling, followed by the remaining half for temporal modeling. This division may lead to the loss of spatial modeling capabilities during subsequent temporal information modeling, ultimately impacting performance. Employing a complete Transformer block might prove more effective in modeling temporal information compared to using multiple attention heads (Variant 3). The detailed FVD values can be seen in Appx. A.5.

**Conditional information injection.** As depicted in Fig 5 (f), the performance of *S-AdaLN* is significantly better than that of *all tokens*. We believe this discrepancy may stem from the fact that *all tokens* only introduces timesteps or label information to the input layer of the model, which could face challenges in propagating effectively throughout the model. In contrast, *S-AdaLN* encodes timestep or label information into the model in an AdalN-like manner for each Transformer block. This information transmission approach appears more efficient, likely contributing to its superior performance and faster model convergence.

**Temporal positional embedding.** Fig. 5 (b) illustrates the impact of two different temporal position embedding methods on the performance of the model. Employing the absolute position embedding approach tends to yield slightly better results than the alternative method.

**Learning with pre-trained models/image-video joint training.** As illustrated in Fig. 5 (c), the initial stages of training benefit greatly from the model's pre-training on ImageNet, enabling rapid achievement of good performance on the video dataset. However, this pre-trained model can pose challenges when attempting to enhance the model's performance further. In other words, as the number of iterations increases, the model's performance tends to stabilize around a certain level. This phenomenon could be attributed to the fact that the pre-trained model is initialized near a local optimum, which allows it to achieve good performance initially. However, breaking away from this local optimal can be challenging. As demonstrated in Tab. 2 and Tab. 1, image-video joint training ("LAVITA+IMG") leads to a significant improvement in the video and image metrics. Including additional independent video frames allows the model to accommodate more examples within each batch (Ho et al., 2022).

| Method | IS ↑ | FID ↓ |
|---|---|---|
| MoCoGAN | 10.09 | 23.97 |
| MoCoGAN-Style | 15.26 | 10.82 |
| VideoGPT | 12.61 | 22.7 |
| MoCoGAN-HD | 23.39 | 7.12 |
| DIGAN | 23.16 | 19.1 |
| StyleGAN-V | 23.94 | 9.445 |
| PVDM | 20.55 | 41.74 |
| LAVITA (ours) | 68.53 | 5.02 |
| LAVITA+IMG (ours) | **73.31** | **3.87** |

Table 1: Inception Score and FID comparisons of LAVITA against other state-of-the-art on the UCF101 and FaceForensics datasets, respectively.

**Video sampling interval.** We collected 16 frames by employing different video sampling intervals. As illustrated in Fig. 5 (a), there is a significant performance gap among the models in the early stages of training. However, as the number of training iterations increases, model performance gradually becomes consistent. We thus chose a video sampling interval of 3 to ensure a reasonable level of continuity in the generated videos.

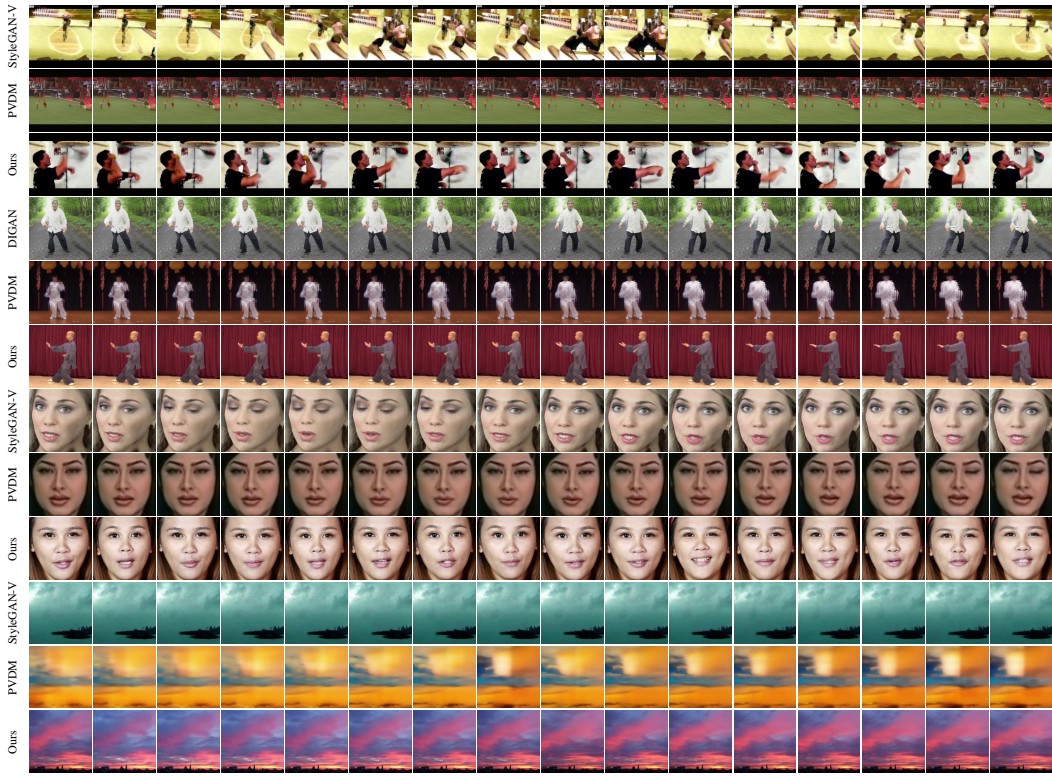

Figure 4: Sample videos from the existing methods on UCF101, Taichi-HD, FaceForensics and SkyTimelapse, respectively.

## 4.3 COMPARISON TO STATE-OF-THE-ARTS

Based on the ablation studies in Sec. 4.2, we conduct a comparison with the current state-of-the-art using Variant 2, uniform frame patch embedding, *S-AdaLN*, and the absolute position embedding approach.

**Qualitative results.** Fig. 4 illustrates the video synthesis results from LAVITA on UCF101, Taichi-HD, FaceForensics and SkyTimelapse. Our method consistently delivers realistic, high-resolution video generation results (256x256 pixels) in both scenarios. This encompasses capturing the motion of human faces and handling the significant transitions of athletes. Notably, our approach excels at synthesizing high-quality videos within the challenging UCF101 dataset, a task where other comparative methods often falter. More results can be seen in the supplementary material.

**Quantitative results.** In Tab. 2, we provide the quantitative results of LAVITA and other comparative methods, respectively. Our method significantly outperforms the previous works on all datasets, which shows the superiority of our method on video generation. In Fig. 1, we report the FID scores on FaceForensics and the Inception Score on UCF101 to evaluate video frame quality. Our method demonstrates outstanding performance with an FID value of 3.87 and an Incetion Score value of 73.31, significantly surpassing the capabilities of other comparative methods.

## 4.4 TEXT-TO-VIDEO GENERATION

We present the results of text-to-video generation on Webvid2M (Bain et al., 2021) and a subset of Laion5B (Schuhmann et al., 2022), comprising approximately 6,400,000 images. To convert discrete text into embeddings, we utilize the identical CLIP text encoder (Radford et al., 2021) employed in Stable Diffusion 1.4. In this context, texts play the same role as the timesteps or labels, leading us to select the pooled output of CLIP. The generated videos are shown in Fig. 6. More results can be seen in the supplementary material. Furthermore, we select 2,048 sampled videos for calculating FVD and FID scores. The resulting FVD and FID values are 378.20 and 47.72, respectively.

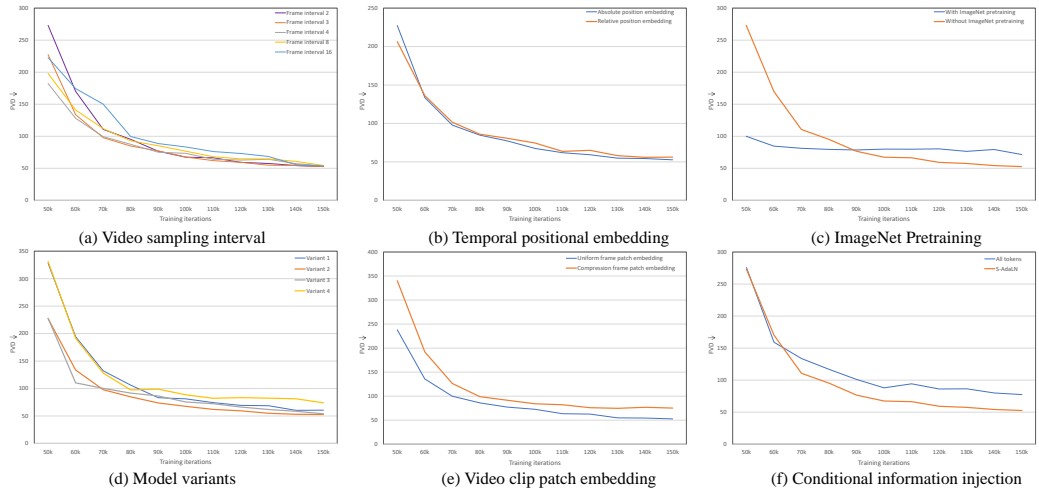

Figure 5: **Ablation of design choices**. We design several ablation studies to explore *best practices* in Transformer-based diffusion models.

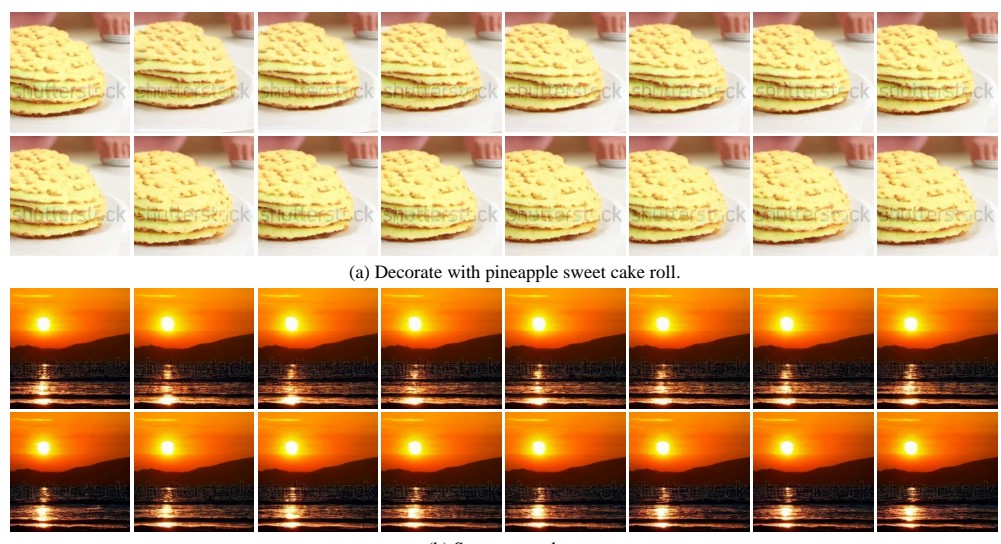

(a) Decorate with pineapple sweet cake roll.

(b) Sunset over the sea.

Figure 6: **Text-conditioned video samples**. LAVITA can generate realistic videos that align with the textual description.

## 5 DISCUSSION

Given the promising results of LAVITA, future work should consider scaling LAVITA to a larger-scale dataset. In our current approach, we treat texts as playing a role akin to labels or timesteps. Thus, we select the CLIP pooled output as the embedding for texts. This inevitably leads to the loss of a considerable amount of text information. Hence, in the future, we will explore alternative methods for effectively fusing text and image information.

## 6 CONCLUSION

This work presents LAVITA, a simple and general latent video diffusion model incorporating sptio-temporal Transformers for video generation. We ablate our main design choices comprehensively and achieve state-of-the-art results across several standard video generation benchmarks and comparable text-to-video generation results. We have a strong belief that LAVITA can offer valuable insights for future research concerning the integration of backbones into diffusion models for video generation.

## ETHIC STATEMENT

We acknowledge the ethical concerns that are shared with other generative models. We aim to synthesize high-quality videos, which can be used for artistic creation and generating synthetic data for other computer vision tasks, etc. We note that our framework has the potential to introduce unintended bias as a result of the training data.

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

# A APPENDIX

## A.1 MODEL PARAMETERS AND FLOPS.

We employed a latent code with an input shape of $(3 \times 16 \times 4 \times 32 \times 32)$ as input for all four models and computed their respective parameters and floating-point operations (FLOPs). The results can be found in Table 3. All model variants have the same complexity of $\mathcal{O}((n_h \cdot n_w)^2 + n_t^2)$.

|  | Variant 1 | Variant 2 | Variant 3 | Variant 4 |
|---|---|---|---|---|
| Params (M) | 673.68 | 673.68 | 676.33 | 676.44 |
| FLOPs (G) | 5572.69 | 5572.69 | 6153.15 | 1545.15 |

Table 3: The number of parameters and FLOPs (Floating-Point Operations) for different model variants

## A.2 THE ARCHITECTURE OF *S-AdaLN* AND VANILLA TRANSFORMER BLOCK.

As shown in Fig. 7, we here present the architecture of *S-AdaLN* described in Sec. 3.3 and the vanilla transformer block used in Model 1 and Model 2.

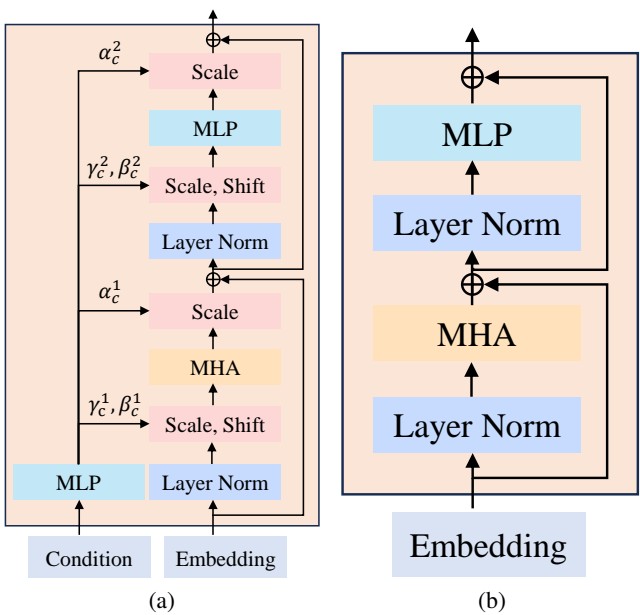

Figure 7: (a) The architecture of S-AdaLN described in Sec. 3.3. (b) The architecture of vanilla transformer block used in Model1 and Model2.

## A.3 MODEL ARCHITECTURE DETAILS.

We present the specifications of Variant 1 as described in Sec. 3. It encompasses a total of 28 transformer blocks, with each block featuring 16 attention heads and a latent size of 1,152 for each attention head. The flops of Variant 1 can be seen in Table. A.1.

## A.4 SAMPLED IMAGES FROM LAVITA WITHOUT THE TEMPORAL TRANSFORMER BLOCKS.

We observe that during inference, the removal of temporal Transformer blocks can directly generate images that align with the textual description. As shown in Fig. 8, we present several sampled images from LAVITA without the temporal transformer blocks on Webv2m and the subset of Laion5B.

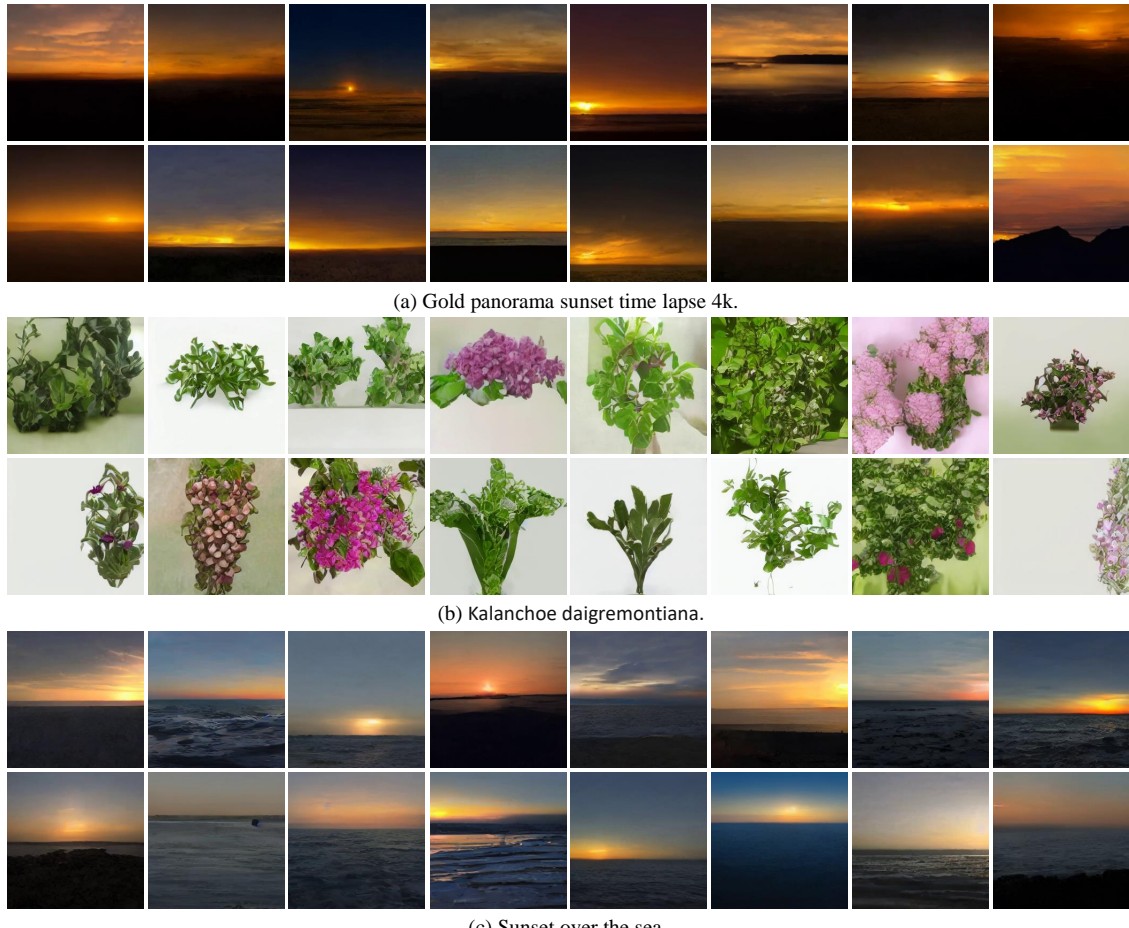

(a) Gold panorama sunset time lapse 4k.

(b) Kalanchoe daigremontiana.

(c) Sunset over the sea.

Figure 8: Sampled images $256 \times 256$ from LAVITA without the temporal transformer blocks.

## A.5 THE DETAILED FVD VALUES OF LAVITA'S FOUR VARIANTS.

The detailed FVD values for the four LAVITA variants utilized in Fig. 5(d) on the FaceForensics dataset are presented in Table 4.

| Iters | Variant 1 | Variant 2 | Variant 3 | Variant 4 |
|-------|-----------|-----------|-----------|-----------|
| 50k   | 328.16    | 227.32    | 227.84    | 331.27    |
| 60k   | 194.33    | 133.66    | 110.32    | 191.53    |
| 70k   | 132.22    | 97.84     | 100.27    | 128.13    |
| 80k   | 106.09    | 84.78     | 91.67     | 97.38     |
| 90k   | 83.18     | 73.69     | 86.41     | 98.79     |
| 100k  | 81.12     | 67.50     | 75.42     | 88.53     |
| 110k  | 74.26     | 61.93     | 72.30     | 82.15     |
| 120k  | 69.20     | 59.31     | 66.21     | 83.29     |
| 130k  | 68.66     | 54.86     | 61.75     | 82.45     |
| 140k  | 60.26     | 53.10     | 58.56     | 81.25     |
| 150k  | 60.51     | 52.73     | 54.03     | 73.85     |

Table 4: The FVD values of four LAVITA variants on FaceForensics used in Fig. 5 (d).

A.6  THE DIFFERENCES BETWEEN LAVITA AND VIVIT.

The architecture and focused task are the main differences between LAVITA and ViViT (Arnab et al., 2021). ViViT is designed to employ transformer-based models for video classification, whereas our LAVITA is focused on employing transformer-based models within latent diffusion models for video generation. Notably, ViViT exclusively relies on classification (CLS) tokens generated by spatial transformer blocks for temporal information modeling, whereas LAVITA avoids the need for an additional CLS token and uses all tokens for spatio-temporal modeling.

