# OpenReview forum: "LAVITA: Latent Video Diffusion Models with Spatio-temporal Transformers"
_ICLR.cc/2024/Conference — ICLR 2024 Conference Withdrawn Submission_

### Official Review · Reviewer_rGPr · 2023-10-26

**Soundness:** 2 fair
**Presentation:** 3 good
**Contribution:** 3 good
**Rating:** 5
**Confidence:** 4

**Summary:**

The paper presents a novel technique called LAVITA with embedding Layer Norm (LN) into neural networks for video generation. The authors demonstrate that the proposed architecture can achieve improved training efficiency and better generalization performance. Experimental results on various datasets and architectures show the effectiveness of the proposed method.

**Strengths:**

1. The performance of the proposed  LAVITA shows a significant improvement compared to previous GAN-based video generation methods (and one diffusion-based method  PVDM)
2. The paper is well-written and clearly organized.

**Weaknesses:**

1.The novelty of this paper is limited and the contribution is incremental. The model design (include the 4 variants, cf. Figure 2), and the approaches for video patch embedding (cf. Figure 3) are directly borrowed from ViViT [1]. The prediction format of the proposed LAVITA (i.e., the Linear layer followed by reshape operation, and the Noise & Variance) is borrowed from DiT [2]. The only difference lies in that here the video input and the noise (as well as the variance) output is in pixel space, while in DiT they build it on latent space.
2.The diffusion-denoising pipeline in this paper is actually in pixel space, e.g., the output noise and variance is of shape FxHxWxC (so the so-called latent video diffusion model in the paper’s title is confusing).
-This design choice contradicts modern diffusion models like Stable-Diffusion, as well as DiT [2]. More analysis and comparisons should be added to explain why the author(s) choose to sampling in pixel space. For example, how about the computational cost or training resources compared to DiT (since DiT sampling in latent space) ?
-This design makes the model rely on extra temporal unsampling module (i.e., the 3D conv module in Sec 3.2) when the input video is tokenized by “Compression frame patch embedding” (cf. Figure 3(b)). The experiment results in Figure 5(e) also shows this makes the model has worse performance than using uniform frame path embedding.

3.The comparative experiments in this paper are not comprehensive. The compared methods are mostly GAN-based, and some are relatively outdated models (only one diffusion-based model. i.e., PVDM[3] is compared). Since the design of LAVITA is highly related with DiT [2], the comparison with DoT is crucial, which however is not presented in this paper. There are also other diffusion-based video generation models which are not discussed/compared in this paper, e.g., [4,5]


[1] ViViT: A Video Vision Transformer, ICCV 2021
[2] Scalable Diffusion Models with Transformers ICCV 2023
[3]  Video probabilistic diffusion models in projected latent space CVPR 2023
[4] Latent Video Diffusion Models for High-Fidelity Long Video Generation, https://arxiv.org/abs/2211.13221
[5] MCVD: Masked Conditional Video Diffusion for Prediction, Generation, and Interpolation NeurIPS 2022

**Questions:**

See weakness.

---

### Official Review · Reviewer_xLU2 · 2023-11-01

**Soundness:** 3 good
**Presentation:** 4 excellent
**Contribution:** 2 fair
**Rating:** 5
**Confidence:** 4

**Summary:**

The paper introduces LAVITA, a video generation architecture that integrates the Transformer architecture into diffusion models. The paper emphasizes the need for effective modeling of spatio-temporal information due to the high-dimensional nature of video data. To address this, the authors proposed to model spatial and temporal information separately, aiming to manage their inherent disparities and reduce computational complexity.
To decide on the optimal architecture, the paper introduces four distinct variants that range from separately processing spatial and temporal information to decompositions of multi-head attention mechanisms within Transformer blocks.
In terms of embedding methods, the paper proposed two frame patch embedding techniques: uniform frame patch embedding and compression frame patch embedding. Additionally, the paper highlights the advantages of image-video joint training, which further increases LAVITA's performance.

**Strengths:**

- The paper is well-written and has a good structure.
- The paper integrates spatio-temporal transformer blocks into the diffusion model to increase the quality of video generation. Their experiments show that separating spatial and temporal blocks improves the video generation performance in comparison to the baseline.
- The experiments on image-video joint training provide valuable insights for video generation research.
- The paper studies two different frame patch embedding methods and conducts experiments on them. This direction seems to have a good potential to explore more.

**Weaknesses:**

The authors explore four architecture variants to capture spatio-temporal information, as well as frame patch embedding methods to embed frames. While the research offers valuable insights, the paper's overall contribution is limited. There are areas that could benefit from further clarification and elaboration:

- In Equation 1, the distribution of "t" is not specified. It would be helpful to include this information.
- The sentence "utilizing a top-down design strategy that spans from the broader perspective of Transformer blocks to the finer details of multi-head attention mechanisms within Transformer blocks.” lacks clarity. consider rewriting in a more clear way.
- Is not very clear how variant 3 is different from variant 2 conceptually.
- In Variant 4, while the authors mention decomposing the MHA into two components, Figure 2(d) presents two MHAs without depicting the decomposition. Could this be clarified?
- In Fig. 2(d) why after the fusion step there is only “z_s”?
- In Variant 4, the term "component" in the statement "We employ different components to attend over tokens along..." is somewhat ambiguous. To which specific components is this referring?
- In the video clip patch embedding, “F” and “s” need to be defined.
- Fig. 5(c) shows that the model without imagenet pretraining obtains better FVD while Tab. 2 shows that the model with imagenet pretraining achieves better FVD. The result doesn’t seem consistent.
- A comparison to the papers [1, 2, 3, 4] is missing.
- The IS number shown in Tab .1 doesn’t match to the IS number in PVDM paper. what is the source of difference?
- some technical information is missing such as training time, the number of GPUs used for training, and the specific type of GPU. This info would be beneficial for reproducibility purposes.
- Some details are missing such as details on the sampling process, including the time required to generate 1 second of video and the number of frames sampled from each video.
- Is it possible to provide a qualitative comparison of different architecture variants?

Ref:

1- Generating Long Videos of Dynamic Scenes, Brooks et al. 2022

2- Align your Latents: High-Resolution Video Synthesis with Latent Diffusion Models, Blattmann et al. 2023

3- Latent Video Diffusion Models for High-Fidelity Long Video Generation, He et al. 2023

4- MoStGAN-V: Video Generation with Temporal Motion Styles, Shen et al.  2023.

**Questions:**

Mentioned in the main review

---

### Official Review · Reviewer_u898 · 2023-11-08

**Soundness:** 2 fair
**Presentation:** 2 fair
**Contribution:** 1 poor
**Rating:** 3
**Confidence:** 3

**Summary:**

Overall, this paper claims that, they integrate the transformer structure into diffusion model for the first time in the realm of video generation. Moreover, they also study how to well design the transformer structure for the diffusion process.

**Strengths:**

1. This paper tested various structures for transformer on different datasets for diffusion, which can be technically valuable for the community.
2. The problem of how to well use transformer structure in diffusion can be an important problem.

**Weaknesses:**

(See the questions section for details)

**Questions:**

Overall, despite the strengthes of this paper, I believe that the current version of this paper is not enough to be accepted. Below are my concerns.
1. The main concern I have on this paper is that it seems misclaim its novelty. Specifically, this paper seems to be emphasize that it pioneer exploring transformer diffusion in video generation. However, after a quick seach online, this seems to be wrong (e.g., VDT: An Empirical Study on Video Diffusion with Transformers).
2. Even worse, after a quick look through of the VDT paper, its designs and explorations seem to be also related to spatial-temporal. While I am not superised on the exploration of spatial-temporal, this seems to leave the submission with limited novelty. Note that the spatial-temporal design has also been explored in other diffusion papers (e.g., DiffPose: SpatioTemporal Diffusion Model for Video-Based Human Pose Estimation).
3. Moreover, I am confused on the qualitative results of this paper as well. From my understanding, as a video generation paper, it is important to show the advantage of the method through the generated videos. However, along the video frames shown in Figure 4 for example, I cannot straightforwardly get the advantage of this method. This leads me to doubt, whether a "best practice" or a "good practice" of transformer in video diffusion is really successfully found.

---

### Official Review · Reviewer_dGeE · 2023-11-10

**Soundness:** 1 poor
**Presentation:** 2 fair
**Contribution:** 1 poor
**Rating:** 3
**Confidence:** 5

**Summary:**

This paper explores several Transformer architecture designs for video diffusion models. To alleviate the high-dimensionality of videos, the proposed method (LAVITA) proposes to model spatial and temporal information separately. They propose to use pretrained DiT weights learned from image datasets as a spatial block initialization.

**Strengths:**

- The paper is generally well-written.
- Since architecture for video diffusion models is still an open problem, the paper targets an important problem.

**Weaknesses:**

- Limited Novelty: There already exists a paper called VDT [Lu et al., 2023] that explore Transformer-based video diffusion model. But the paper misses any discussions and comparing results with this method.
- Motivation: I think the main advantage of exploring Transformer architecture for diffusion models is in its good scalability compared with popular U-Net based architectures, as shown in the recent DiT paper [Peebles & Xie, 2023] that extensively shows the correlation between performance vs. # of parameters / computations. However, this paper misses such analysis and even worse, the performance is not that satisfactory, underperforming previous work (see Evaluation below)
- Evaluation: The paper argues the proposed LAVITA shows the state-of-the-art performance, but this it NOT true. Specifically, the values of PVDM in Table 2 and 3 is really weird. For instance, on UCF-101, the FVD score of PVDM is less than 400, outperforming 800 of LAVITA, but they are reported as significantly worse values (2000) in this Table 2. Considering this paper releases official implementation, checkpoints, these authors still tries their best to address the problem, I have a strong doubt on the evaluation results and suspect authors no not faithfully train this baseline. In this respect, I cannot believe the authors' evaluation, and considering the original FVD of PVDM is below 400 on UCF-101, it is difficult to say LAVITA is a state-of-the-art video generation method.

**Questions:**

See my weakness